# Relevance of Host Cell Surface Glycan Structure for Cell Specificity of Influenza A Viruses

**DOI:** 10.3390/v15071507

**Published:** 2023-07-05

**Authors:** Markus Kastner, Andreas Karner, Rong Zhu, Qiang Huang, Andreas Geissner, Anne Sadewasser, Markus Lesch, Xenia Wörmann, Alexander Karlas, Peter H. Seeberger, Thorsten Wolff, Peter Hinterdorfer, Andreas Herrmann, Christian Sieben

**Affiliations:** 1Institute for Biophysics, Johannes Kepler University Linz, 4020 Linz, Austria; 2State Key Laboratory of Genetic Engineering, Shanghai Engineering Research Center of Industrial Microorganisms, MOE Engineering Research Center of Gene Technology, School of Life Sciences, Fudan University, Shanghai 200438, China; 3Department for Biomolecular Systems, Max Planck Institute for Colloids and Interfaces, 14476 Potsdam, Germany; 4Institute of Chemistry and Biochemistry, Freie Universität Berlin, 14195 Berlin, Germany; 5Division of Influenza and other Respiratory Viruses, Robert Koch-Institute, 13353 Berlin, Germany; 6Molecular Biology Department, Max Planck Institute for Infection Biology, 10117 Berlin, Germany; 7Institut für Chemie und Biochemie, Freie Universität Berlin, Altensteinstraße 23a, 14195 Berlin, Germany; 8Nanoscale Infection Biology Group, Department of Cell Biology, Helmholtz Centre for Infection Research, 38124 Braunschweig, Germany; 9Institute for Genetics, Technische Universität Braunschweig, 38106 Braunschweig, Germany

**Keywords:** influenza A virus, cell binding, force spectroscopy

## Abstract

Influenza A viruses (IAVs) initiate infection via binding of the viral hemagglutinin (HA) to sialylated glycans on host cells. HA’s receptor specificity towards individual glycans is well studied and clearly critical for virus infection, but the contribution of the highly heterogeneous and complex glycocalyx to virus–cell adhesion remains elusive. Here, we use two complementary methods, glycan arrays and single-virus force spectroscopy (SVFS), to compare influenza virus receptor specificity with virus binding to live cells. Unexpectedly, we found that HA’s receptor binding preference does not necessarily reflect virus–cell specificity. We propose SVFS as a tool to elucidate the cell binding preference of IAVs, thereby including the complex environment of sialylated receptors within the plasma membrane of living cells.

## 1. Introduction

Influenza A viruses (IAVs) circulate in aquatic birds, their natural host reservoir, but have also established stable lineages in various mammalian species such as pigs. Although animal IAVs usually remain confined within their natural host species, they can cause zoonotic infections in humans on rare occasions [1]. Such interspecies transmissions can result in clinically severe or even fatal respiratory disease in humans, as illustrated by the outbreaks of avian-origin H7N9 subtype viruses in China occurring since 2013 [2]. Zoonotic transmission events can, in fact, largely influence the epidemiology of human influenza directly if the virus succeeds to spread efficiently among humans, as was observed in 2009 for the pandemic swine-origin H1N1 strain (pdmH1N1) [3]. Although the genetic requirements for crossing the species barrier are still incompletely understood, it is accepted that interspecies transmission of IAVs at least partially depends on the capability of the spike protein hemagglutinin (HA) to recognize specific sialylated glycan receptors on the host cell surface [4]. In general, the HA of avian viruses preferentially binds to α-2,3-linked sialic acid (SA) (avian-type receptor), whereas the HA of human-adapted strains strongly binds to terminal α-2,6-linked SA (human-type receptor) [5]. Several studies have determined that alterations in HA’s receptor specificity are a crucial step in host adaptation and interspecies transmission for several IAV subtypes [4]. However, it remains speculative if those adaptive mutations result in increased virus–cell binding.

Glycan arrays containing a variety of synthetic glycans are widely used to characterize IAV receptor specificity with a high precision of structural glycan properties. Recent studies of the cellular glycome of the human and swine respiratory tract tissues show that its complexity might not be adequately represented by current glycan arrays [6,7,8,9]. Consistently, IAVs were shown to use a more diverse group of glycans [7,10] or even sialic-acid-independent attachment factors [11]. Candidate molecules include C-type lectins (L-SIGN and DC-SIGN) that were found to participate in influenza virus attachment independent of SA specificity [12]. Hence, complementary approaches to directly assess viral receptor specificity within the complex environment of the cell surface are necessary to reach a more comprehensive understanding of the initial stage of virus infection. Atomic force microscopy (AFM)-based single-virus force spectroscopy (SVFS) allows for measuring the binding of individual IAVs to living host cells at the molecular level [13,14,15]. In this type of analysis, intact viruses are covalently attached to AFM cantilevers. Cycles between the cantilever–cell approach, cell binding, and cantilever retraction allow direct characterization of virus–cell binding while revealing kinetic and thermodynamic properties of the interactions. Thus, SVFS allows for investigating virus–cell binding in an experimental system that closely mimics the natural situation [16,17].

Using five different IAV strains (Table 1), we systematically address whether virus receptor specificity observed by glycan array analysis is reflected in virus–cell binding patterns as determined by SVFS. Our results indicate that data obtained from in vitro glycan arrays may not directly reflect virus–cell binding specificity. We suggest that host cell specificity does not solely depend on the sialic acid configuration of the cell surface but is more complex and depends on the specific environment of the receptor and possibly involves additional attachment factors or co-receptors with yet unknown functional roles.

## 2. Materials and Methods

### 2.1. Cell and Virus Propagation

Chinese hamster ovary (CHO) cells and human alveolar A549 cells were grown in DMEM (PAA) supplemented with 1% penicillin/streptomycin and 10% FCS (PAA) in plastic Petri dishes. For sialic acid digestion, we used neuraminidase (NA) from *Clostridium perfringens* (Sigma-Aldrich, St. Louis, MI, USA) solved in PBS buffer. The cells were treated for 10 min at 37 °C with 1 U/mL NA. Influenza A viruses were grown on 10-day-old chicken eggs and purified from allantoic fluid by gradient centrifugation through a 20–60% (*w*/*v*) sucrose gradient. The A/Anhui/1/2013 strain was inactivated by UV irradiation before gradient centrifugation. Virus strains AH1 and pdmH1N1 were taken from the strain collection at RKI, and FPV was provided by Dr. Michael Veit (Free University Berlin).

### 2.2. Glycan Array

Glycan array preparation was performed as described previously [18]. Briefly, glycans containing a primary amino linker were dissolved at a concentration of 0.1 mM in printing buffer (50 mM sodium phosphate, pH 8.5) and printed on N-hydroxysuccinimide activated glass slides (CodeLink slides, Surmodics, Edina, MN, USA) using an S3 robotic microarray spotter (Scienion, Berlin, Germany). Slides were incubated overnight in a humidity-saturated chamber and remaining reactive groups were quenched by incubating with 100 mM ethanolamine and 50 mM sodium phosphate at pH 9.0 for 1 h at room temperature. Slides were washed with water, dried by centrifugation, and stored at 4 °C until use. Before loading, the array was washed with DPBS. Virus was diluted as indicated into sterile binding buffer containing 1% BSA, 0.05% Tween 20 (MERCK), CaCl_2_ (492 µM), and MgCl_2_ (901 µM) at pH 7.0. Then, 30 µL of diluted virus was pipetted in each well and the array was incubated in a moist chamber for 24 h at 4 °C. Each well was then washed three times with washing buffer containing DPBS and 0.1% Tween 20 (DPBS-T). Subsequently, wells were blocked with DPBS containing 1% BSA for 2 h at 4 °C and permeabilized using DPBS-T containing 0.3% Triton-X100. To stain the bound virus the array was incubated with a primary monoclonal antibody against the viral NP protein (1:1000, clone AA5H, AbD Serotec, Oxford, UK) at 4 °C overnight. Primary antibody was removed, and wells were washed three times with DPBS-T. Secondary Cy3-coupled goat anti-mouse IgG (1:100, product-code: 115-165-146, Jackson ImmunoResearch Laboratories, West Grove, PA, USA) was added and incubated at RT for 1 h. The array was washed three times with DPBS-T and dipped into distilled water before scanning. Glycan array fluorescence images were obtained using a GenePix 4300A microarray scanner (Molecular Devices, Sunnyvale, CA, USA). Fluorescence intensities of spots were evaluated with GenePix Pro 7.2 (Molecular Devices).

### 2.3. AFM Tip Chemistry

Commercially available AFM cantilevers (MSCT, Bruker) were amine functionalized by using the room-temperature method for reaction with APTES^13^. A heterobifunctional PEG linker, acetal–PEG_800_–NHS (N-hydroxysuccinimide) (Figure 2A), was attached by incubating the tip for 1.5–2 h in 0.5 mL of chloroform containing 2 mg/mL acetal–PEG–NHS and 8 µL triethylamine, resulting in acylation of surface-linked APTES by the NHS group. The terminal acetal group was converted into an amine-reactive aldehyde by incubation in 1% citric acid as described previously [13]. After rinsing with water 3 times, once with ethanol, and drying under a stream of nitrogen, the tips were incubated in a mixture of 19–25 μL of approximately 0.6–1.6 mg/mL influenza A virus in PBS (without Ca^++^) and 1–2 μL of 1 M NaCNBH_3_ (freshly prepared by dissolving 32 mg of solid NaCNBH_3_ in 500 μL of 10 mM NaOH) for 60 min. The tips were then washed in 3 mL PBS 3 times and stored in PBS at 4 °C. All other chemicals and reagents were purchased from different commercial sources in the highest purity grade available. We want to point out that at the end the entire AFM cantilever was coated with IAV particles and that the shape of the AFM cantilever with its sharp tip eventually allows single virus interaction to be measured.

### 2.4. SVFS Measurement

AFM-based force spectroscopy was performed at room temperature with an Agilent 5500 AFM. The Petri dish with cells was mounted with the AFM, which was put on the optical microscope through a specially designed XY stage. Before force measurements, the cantilever with a nominal spring constant of 10 pN/m functionalized with influenza A virus was incubated in 5 mg/mL BSA for 30 min in order to minimize the nonspecific interaction between the cantilever tip and the cell surface. Measurements were performed in PBS buffer at room temperature. After the cantilever tip approached the cell surface, force–distance curves were repeatedly measured with Z-scanning range of 2 μm, cycle duration of 0.5–8 s, 500 data points per curve, and typical force limit of about 40–70 pN. The spring constants of the cantilevers were determined by using the thermal noise method [19].

### 2.5. Fitting of SVFS Data

Similar to single-molecule force spectroscopy (SMFS), also in SVFS studies, several hundred force–distance cycles are recorded in a dynamic range of increasing loading rates under identical conditions. For each of these force curves showing unbinding events, the unbinding force Fi and the effective spring constant $k_{eff}$ (slope at rupture) were determined. The loading rates r were determined by multiplying the pulling velocity v with the effective spring constant $k_{eff}$ (i.e., $r=v*k_{eff}$). Additionally, a rupture force probability density function (pdf) (Figure 3B) was calculated, and a Gaussian distribution was fitted to the main peak of the pdf. Subsequently, all unbinding events within μ±σ of the fit were selected to create a loading rate dependence scatter plot (Figures 3C and S2) for further calculations of $k_{off}$ and $x_{u}$.

Generally, the loading rate r is constant for a fixed pulling speed, which implies that the effective spring constant $k_{eff}$ does not vary significantly. However, for force spectroscopy measurements on live cells it is known that $k_{eff}$ could show a broadened distribution caused by local variations in the spring constant of the cell surface, leading to a convolution of the rupture force distribution and further influencing the calculations for the dissociation rate constant, $k_{off}\;$, and the separation of the receptor-bound state to the energy barrier, $x_{u}$. To circumvent this influence, we applied a maximum likelihood routine to fit the SVFS data to the Evans model [20] in order to obtain $k_{off}$ and $x_{u}\;$(Table 2).

According to the single energy barrier binding model, the probability *p* that the complex breaks at a certain force, *F*, is given as [21]:(1)$$p\left(F\right)=\frac{k_{off}}{r}exp\left[\frac{F_{xu}}{k_{B}T}-\frac{k_{off}k_{B}T}{rx_{u}}\left(exp\frac{Fx_{u}}{k_{B}T}-1\right)\right]\;\;$$

The parameters $x_{u}$ and $k_{off}$ were determined by applying a maximum likelihood approach, in which the negative log likelihood *nll* was minimized by modifying $k_{off}$ and $x_{u}$, with p based on Equation (1) defined in the single barrier model [21]:(2)$$nll=-\underset{t}{\sum }\log p\left(k_{off},x_{u},F_{t},r_{t}\right)$$

We want to point out that the model from Friddle and De Yoreo [22] is a more recent formalism that could be used as an alternative to the single energy barrier binding model used here. For reasons of comparability with our previous SVFS work on live cells, we used the Bell–Evans model for all our data in this study.

## 3. Results

To investigate and compare the SA receptor specificity of the different IAV strains, we performed an in vitro glycan array study using a library of 15 selected glycans that are typically found in mammalian cells (Figure 1). Regarding specific IAV receptors, our library included three α-2,3-linked (avian-type) SA conjugates as well as three α-2,6-linked (human-type) SA conjugates (grouped on the left side in Figure 1). Sialyl-Lewis^X^ (SLe^X^, glycan number nine), although terminally linked to α-2,3 SA, has a different topology due to its fucosylation and thus is shown separately.

We found that the zoonotic human H7N9 isolate A/Anhui/1/2013 (AH1) and the pandemic H1N1 (pdmH1N1) virus recognized all six SA conjugates. While pdmH1N1 bound the α-2,6 conjugates equally well, it showed reduced binding to the α-2,3 conjugates, overall suggesting a preference of α-2,6 conjugated sialic acid. While a dual binding behavior was already observed before for pdmH1N1 (A/California/04/2009 and A/Hamburg/5/2009) [23,24], the receptor binding preference of the AH1 isolate is still under debate. It was shown that AH1 exhibits increased human receptor binding while still preferring avian receptors [25], and others reported on the specificity for human-type receptors [26]. The avian H7N1 isolate A/FPV/Rostock/1934 (FPV) recognized all three avian-type conjugates, but bound only one human-type conjugate, which is in line with previous findings using glycan arrays [27].

We further tested H3N2/X31 as well as H1N1/WSN. H3N2/X31 carries the HA of the human pathogenic IAV strain A/Aichi/68, which was also previously shown to prefer α-2,6-linked (human-type) receptors [28]. In line with that, we found that H3N2/X31 only recognized human-type SA conjugates on the glycan array. The lab-adapted H1N1/WSN was previously shown to prefer α-2,6-linked (human-type) receptors over α-2,3-linked SA on re-sialylated erythrocytes [29]. In our hands, H1N1/WSN bound all six receptors with no obvious preference.

Interestingly, among all glycans, we observed the strongest binding for pdmH1N1, H1N1/WSN, and AH1 to the kinked, fucosylated SLeX (Figure 1). SLeX, mostly known to be involved in leukocyte binding, was identified in human respiratory tract tissue [8] including A549 cells [30] and shown to be recognized by different IAV subtypes [31]. In particular avian H7 IAVs [32]. The avian-origin FPV (not tested in [32]) was shown to recognize only the sulfated form of SLeX [31], which could explain the absence of binding in our experiments. We also included six glycans without terminal sialic acid on our glycan array (Figure 1, right side). While we indeed observed virus binding, it was on average 10–100-fold lower compared to the sialylated glycans, indicating that our tested viruses require terminal sialic acid for improved binding.

We next used SVFS to characterize virus binding to two different model cell lines, A549 and CHO. A549 cells are derived from the lower human respiratory tract and express both major SA receptor types on the cell surface, as shown by lectin binding [14]. CHO cells lack an α-2,6-specific sialyltransferase and only express α-2,3-linked SA [33]. Hence, we chose them as a comparative model for studying viral binding to cells displaying only avian-type receptors [14,18]. The presence of both glycans on the respective cell types was confirmed by fluorescent lectin staining using *Sambuccus nigra* agglutinin (SNA, specific for α-2,6-linked SA) and *Maackia amurensis* agglutinin (MAA, specific for α-2,3-linked SA) (Appendix A). For SVFS, intact viruses were covalently attached to AFM cantilevers using a bifunctional crosslinker (Figure 2A).

To measure virus–cell interaction forces, functionalized cantilevers were lowered on adherent cells until touching the cell surface, then raised again until the formed interaction was broken and detected as a jump of the force vs. distance curve (Figure 2C). Binding to cells was measured in a dynamic range of increasing loading rates *r*, i.e., pulling velocities to determine the dissociation rate at zero force *k_off_*. Unbinding events were recorded and analyzed to obtain the rupture force *F* as well as the effective spring constant *k_eff_*, defined as the slope of the force–distance curve at rupture (Figure 3A). A typical rupture force histogram and the accompanying probability density function (pdf) of the interaction between influenza AH1 and A549 cells are shown in Figure 3B. During this experiment, we detected a binding probability of 29.7% (*v* = 500 nm/s). A typical spherical influenza A virion has approximately 300 HA trimers, i.e., 900 sialic acid binding sites [34]. On the cellular side, SA is a ubiquitous glycan, as we also show by lectin binding (Appendix A). We thus hypothesize a high chance of HA–SA binding during virus–cell attachment. Reducing the SA density on the cell surface should thus only reduce the probability of HA–SA binding but not the overall measured force. After cell surface SA depletion by neuraminidase (NA) treatment, the binding probability was reduced to 5–13%, while the pdf peak position was unchanged (red curves and inset in Figure 3B), indicating the specificity of our measurements. The binned histograms are shown for comparison along with the fitted pdf.

**Figure 3 viruses-15-01507-f003:**
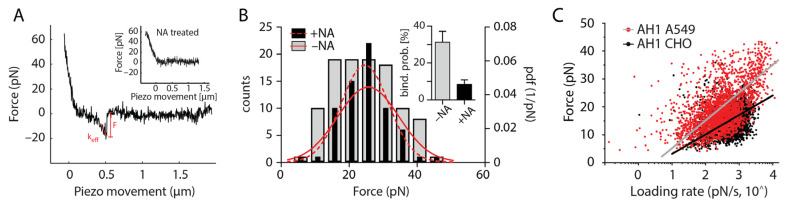
SVFS measurements of H7N9 viruses interacting with receptors on living cells. (**A**) Force trace of H7N9 AH1 virus–cell interaction measured by AFM-based SVFS showing a characteristic single unbinding event. After treating the cells with neuraminidase (NA), the binding probability was strongly decreased (see also inset in **B**), causing a high number of force traces showing no interaction (inset in (**A**)). (**B**) Force histogram (left Y axis) and overlaid force probability density function (pdf, right Y axis) of AH1 virus–A549 cell interaction before and after NA treatment. The observed force values were found to be very similar, but the binding probability was strongly decreased (inset, mean ± SD, *n* = 3). (**C**) Scatter plots showing unbinding force *F* plotted against the loading rate *r* of every individual force curve from AH1 virus–A549 cell interaction. The red dashed line shows the fitting to the single energy barrier model (see Materials and Methods) to retrieve the thermodynamic properties of the interaction (summarized for all virus–cell combinations in Table 2). For each virus–cell combination, we measured between 12–15 cells. All force vs. loading rate scatter plots are shown in Appendix A.

From *k_eff_*, the loading rate *r* (i.e., force per time) was calculated by multiplication with the retraction velocity *v*. Notably, we used an adapted data analysis procedure, which takes the variable local conditions of a living cell surface into account [20]. Briefly, although the loading rate *r* should be constant for a given pulling speed *v*, recent studies have shown that the heterogeneity of a living cell surface leads to a broad distribution of observed loading rates [20]. Hence, to account for this effect, our approach does not rely on binning of loading rates but takes each individual force–distance curve into account as shown for AH1 virus–A549 cell interactions (Figure 3C). By fitting the force spectra to a single energy barrier model (Figure 3C, see Section 2), we obtained the thermodynamic properties of the interaction (summarized in Table 2). The dissociation rate *k_off_* and its reciprocal, the bond lifetime *τ_off_*, provide information about the stability of the underlying virus–cell interaction.

**Table 2 viruses-15-01507-t002:** Dissociation rate *k*_off_, separation from the energy barrier *x*_u_, and average bond lifetime *τ*_off_ obtained by fitting the SVFS data to a single energy barrier binding model as described in Materials and Methods (see also Figure 3D). The mean +/− SD is shown.

Cell (Receptor Type)	*x*_u_ (Å)	*k*_off_ (s^−1^)	τ_off_ (s)
Virus AH1 (H7N9)			
CHO (avian-like)	13.2 ± 0.016	1.17 ± 0.001	0.85
A549 (human-like)	9.24 ± 0.006	0.69 ± 0.0008	1.43
Virus FPV (H7N1)			
CHO (avian-like)	2.40 ± 0.004	0.33 ± 0.001	3.03
A549 (human-like)	5.74 ± 0.016	0.24 ± 0.001	4.15
Virus pdmH1N1			
CHO (avian-like)	12.5 ± 0.004	0.19 ± 0.001	5.20
A549 (human-like)	5.2 ± 0.016	0.2 ± 0.001	5.00
Virus X31 (H3N2)			
CHO (avian-like)	9.54 ± 0.18	0.66 ± 0.05	1.51
A549 (human-like)	6.42 ± 0.09	1.27 ± 0.07	0.78
Virus WSN (H1N1)			
CHO (avian-like)	2.77 ± 0.04	0.62 ± 0.04	1.61
A549 (human-like)	2.67 ± 0.04	0.85 ± 0.05	1.18

For AH1, we observed pronounced binding to both tested cell lines, with rupture forces between 10 and 100 pN depending on the applied loading rate (Figure 3C and Appendix A). However, we found an about 40% reduced dissociation rate for A549 compared to CHO cells, indicating preferential binding of human-type cell surfaces (Table 2). We confirmed this binding preference of AH1 by measuring binding to living MDCK cells, which express, similar to A549 cells, both human- and avian-type receptors (Appendix A). We observed preferential binding to MDCK cells compared to CHO cells. For FPV, we observed about three times lower dissociation rates compared to AH1, with preferential binding to A549 (see Table 2). The pdmH1N1 virus showed similar dissociation rates as FPV, but without pronounced cell type preference. A comparison of the mean unbinding force at 4 µm/s pulling velocity is shown for all virus–cell combinations in Appendix A. While we found the lowest mean unbinding force for AH1, pdmH1N1 and H7N1/FPV show unbinding forces 4–6 times larger, possibly indicating an affinity difference. The high unbinding forces are in agreement with the lower off rates for pdmH1N1 and H7N1/FPV compared to the other virus–cell pairs.

We have studied H3N2/X31 and H1N1/WSN before by SVFS [14] but now reanalyzed our data using the improved fitting procedure described above. Although we found quantitative differences between the two analysis methods, they are in qualitative agreement, leading to the same conclusions. The fitting values are reported in Table 2, and a comparison with the previously used analysis is summarized in Appendix A. H3N2/X31 showed stronger attachment to CHO cells, while binding of H1N1/WSN to A549 and CHO cells was almost identical.

## 4. Discussion

Zoonotic avian to human IAV transmission events are often accompanied by adaptive mutations within the HA spike protein. One feature used to predict the transmissibility of a potentially human-pathogenic IAV is the specificity of HA for human- compared to avian-type sialylated glycans. It is not well established how HA’s receptor specificity affects virus–cell binding. The plasma membrane is a complex environment with a high diversity of glycans and other potential virus attachment molecules that may affect IAVs’ cell specificity.

We have performed a comparative study combining glycan array binding to evaluate the receptor specificity of various IAV strains along with testing their cell specificity using SVFS. While our glycan array results are largely in line with previous findings using similar assays [23,24,25,26,27,28,29,31], the SVFS results (Table 2) suggest that HA’s preference for human- or avian-type receptors does not necessarily correlate with the expected binding patterns to model cell lines mimicking the surfaces of human or avian cells (see below). For pdmH1N1 and H1N1/WSN, we found that the SVFS data (for H1N1/WSN see [14]) are in good agreement with results obtained from glycan array binding, as neither strain displayed a strong preference for human- or avian-type receptors or a particular cell model. However, we observed contradicting preferences for H3N2/X31, AH1, and also for FPV. We found H3N2/X31 to preferentially bind avian-type cell surfaces, while only recognizing α-2,6-linked (human-type) receptors on the glycan array. AH1, similar to pdmH1N1, recognized both receptor types on the glycan array and showed good binding to all six presented specific glycans on the array, while SVFS indicated a preference for human-type cell surfaces (Table 2). FPV recognized all three avian-type receptors but only one human-type receptor on the glycan array, while showing preferential binding to human-type cell surfaces in SFVS. However, binding to the recognized human-type receptors (receptor 3 in Figure 1) was about two- to three-fold stronger when compared to the avian-type receptors (receptor 5 in Figure 1), which might explain the stronger binding to A549 cells. In conclusion, our results suggest that HA’s receptor preference as tested in glycan array binding has to be considered with caution when concluding on the preferred binding to human-type over avian-type cell surfaces.

Our findings described above raise the possibility that non-sialic acid receptors could contribute to a larger than expected extent to virus–cell binding. The concept of non-sialylated receptors being involved in IAV infection is intriguing and under discussion; since the observation of IAV infection in desialylated cells [11] and due to advanced glycomics [7], the idea has received more attention recently. However, SVFS analyses of cells after pre-treatment with neuraminidase to remove sialic acid structures showed that the binding probability was strongly reduced, leaving the unbinding force unchanged (Figure 3B). This indicates that the viruses indeed mainly bind to sialic acid of the cell surface, but that the local organization and environment of the receptors or other cell surface molecules alter the macroscopic cell specificity leading to the observed differences. The nanoscale organization of specific IAV-binding glycans was investigated before using super-resolution microscopy showing compartmentalization at the scale of an individual IAV particle (<100 nm) [35]. Nanoscale imaging together with more specific labeling approaches and refined glycomics will help to better explore how viruses attach to live cells.

The stronger binding of AH1, pdmH1N1, and H1N1/WSN to the fucosylated glycan SLe^X^ in comparison to the other α-2,3-linked (avian-type) SA of our glycan array is indicative for the relevance of the glycan topology, as also found previously [36]. From the viral point of view, cumulating evidence suggests a role of the viral neuraminidase (NA) in contributing to cell binding via sialic acid [37]. Reiter-Scherer et al. recently used AFM-based force spectroscopy to compare sialic acid binding to either NA or HA and found a strong interaction between NA and sialic acid [38]. In our SVFS measurements, NA was kept unperturbed and, hence, it cannot be excluded as a binding mediator, a feature that could be tested in future experiments.

## 5. Conclusions

Recent glycomics approaches and the use of ex vivo tissue culture revealed new insights into the complexity of the living cell surface [7,8]. Since sialic acid was first identified as an influenza virus attachment factor, many studies have focused on HA–SA binding. Although this interaction is important, not only infection of desialylated cells [11] but also the recent characterization of non-SA binding HA encoded by a bat-derived H17N10 virus [39], in addition to the discovery that the 1918 pandemic virus unaffectedly binds to primary human airway cells even when its HA is engineered to bind exclusively to avian-type SA receptors [40], suggested that other molecular determinants within the plasma membrane are also critical in initiating influenza A virus infection. For characterizing virus specificity, we suggest a dual complementary approach: (1) in vitro binding assays with synthetic glycans to precisely identify the preference of HA (or NA) for a specific sialic acid structure and (2) SVFS as demonstrated here to unravel the cell specificity, modulated by the local environment of the living host cell. Notably, fluorescent soluble HA trimers represent an elegant alternative to probe HA specificity to living cells [41]. We have recently demonstrated this complementary approach for an adapted mutant of pdmH1N1 [18]. While glycan array analysis could not identify a switch in receptor preference, SVFS revealed that the adaptive mutation in HA strongly reduced the binding strength without changing the cell specificity. These binding properties are not accessible and might be hidden when only using in vitro specificity assays. The use of new methods such as SVFS [16] and ex vivo tissue culture in combination with global glycomics and proteomics approaches could help to identify essential components of the plasma membrane facilitating influenza virus–cell interaction.

## Figures and Tables

**Figure 1 viruses-15-01507-f001:**
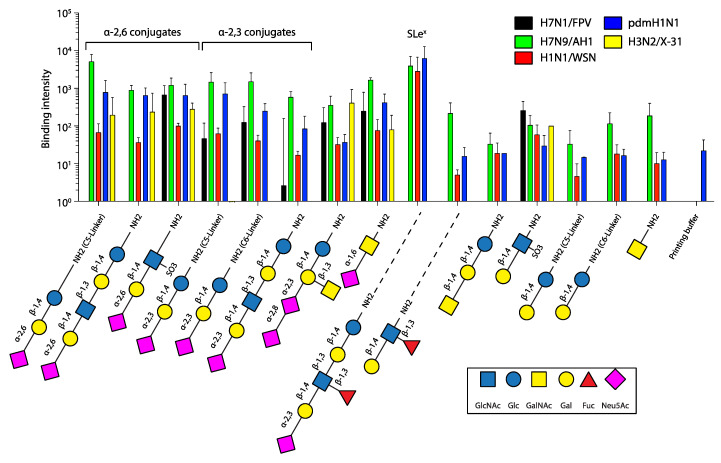
Binding characteristics of influenza A viruses to sialic-acid-conjugated receptors quantified by glycan arrays. Equal amounts of the indicated viruses were bound to glycan arrays, spotted with 15 different sialic acids and as a negative control with printing buffer. Staining of bound viruses was achieved using an NP-specific primary antibody and a Cy3-coupled secondary antibody. The results represent the mean +/− SD for two independent experiments. The disaccharide Sia-α2,6-GalNAc is not grouped with the α-2,6 conjugates on the left due to the different linkage to GalNAc, not Gal. Sialyl-Lewis^X^ (SLe^X^) has a different topology due to its fucosylation and is thus also shown separately. The results represent the mean + SD for two independent experiments.

**Figure 2 viruses-15-01507-f002:**
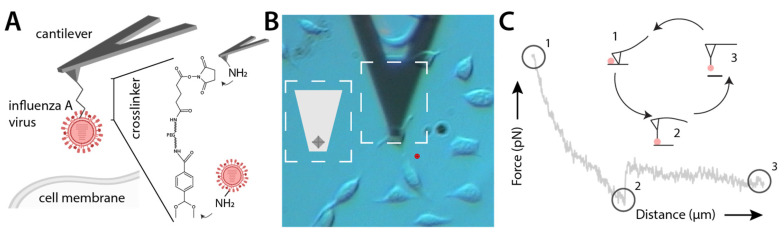
Schematic diagram of the SVFS experimental setup using atomic force microscopy (AFM). (**A**) General principle of AFM-based SVFS. Intact influenza A virions are covalently attached to AFM cantilevers using a bifunctional acetal-PEG_800_-NHS crosslinker. The cantilever can be positioned on top of adherent cells growing in culture dishes (**B**). The combination with light microscopy allows identification of the cantilever with its pyramidal cantilever tip (**B**, inset shows a graphical illustration of the cantilever shown in the respective dotted rectangle) and thereby precise positioning. (**C**) An illustration of a force–distance cycle (top) together with a resulting typical force–distance curve (bottom). During a force–distance cycle (**C**), the cantilever is lowered on a single cell until touching the cell surface (1). Subsequently, the cantilever is retracted at a defined velocity *v*. In the case of an interaction, the cantilever will bend towards the sample (2) until the underlying bond fails and the cantilever returns into the zero-force position (3). The cantilever acts as a Hookean spring and hence bending can be translated into applied force.

**Table 1 viruses-15-01507-t001:** Overview of the influenza A virus strains used in this study.

Virus Strain	Abbreviation	Origin
A/Anhui/1/2013 (H7N9)	AH1	Human, zoonotic, and avian origin
A/FPV/Rostock/1934 (H7N1)	FPV	Avian
A/Hamburg/5/2009 (H1N1)	pdmH1N1	Human, pandemic strain
A/WSN/1933 (H1N1)	WSN	Human
A/X-31 (H3N2)	X31	Human, HA gene from A/Aichi/68 (H3N2)

## Data Availability

Appendix A are provided. Raw data are available upon request.

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
