# Peer review of "Relevance of Host Cell Surface Glycan Structure for Cell Specificity of Influenza A Viruses"

_viruses, 2023, doi:10.3390/v15071507_

Round 1

Reviewer 1 Report

The article by Kastner et al. entitled “Relevance of host cell surface glycan structure for cell specificity of influenza A viruses” combined glycan arrays with single virus force spectroscopy (SVFS) to study and compare binding preferences of several Influenza A virus strains. While this study is likely to be of interest to the readership of the viruses journal especially in view of its unconventional combination of methodologies to study virus binding, publication of the submitted manuscript in its current form appears to be premature.  Please see the comments below.

-          Figure captions are missing for all main figures making the reading currently a bit cumbersome.

-          Additionally, no information is given on the number of repeats, reproducibility, statistical significance, etc of the experiments. This should be stated as relevant.

-          The authors carry out glycan array experiments to investigate the glycan-binding specificity of the different strains and to relate them to virus-cell binding patterns using established cell lines. This is a very interesting idea, however, to fully realize its potential, more detailed information on the carbohydrate profile (especially when it comes to the sialylated residues) of both cell lines should be provided. Relative abundance would also be very interesting. If this is not available from the literature, glycomic data (mass spectrometry?) would enhance the impact of the publication. For example, SLex, seems to be recognized very well by some of the tested strains. Is it present and abundant in both cell lines?

-          Reading would be greatly facilitated if the authors more clearly discussed at the beginning of the result section the different strains studied here and why they were chosen for this study. A table summarizing whether they are zoonotic, pandemic, human, or avian (etc..) would be most useful (Could perhaps be added to Table 1). In particular, it becomes a bit difficult to easily grasp the main conclusions from the SVSF data without this.

-          The sources for the different virus strains are lacking in the materials and methods section.

-          Also reading would be facilitated if the authors detailed how and why the different glycans of the glycan array were selected.  (Now it just says “selected glycans”, line 179, section 3.1.)

-          The authors do not comment on the binding to non-sialylated glycans included in Figure 1 anywhere in the text. They should either do so or not present this data in the figure if they deem it not relevant to the paper.  

-          The sentence “if there is any receptor binding presence of the AH1 Isolate is still under debate”. This statement is unclear. Do the authors mean in the literature or in their own data? Do they mean preference for alpha 2,3 vs. alpha 2,6? Please clarify!

-          Also, the authors state that the pdm H1N1 recognizes all 6 SA conjugates without any distinct preference. However, it appears that binding is less prominent for the second and especially the third (left to right) glycoconjugate, as compared to the first one. Perhaps the statement needs to be refined and this aspect needs to be commented on.

-          Even though the binding probability reduction by about 10% after NA treatment suggests that binding is specific to SA, it would be best to confirm this with an orthogonal measurement, using either a tip without a virus or soluble SA as a binding inhibitor.

-          Did the authors fit their force data with Friddle-De Yoreo model? Data in Figure 3C suggest that there are two single energy barriers and not just one. This is especially visible if we compare this figure with the data provided in Figure S2. How do other Dynamic Force Spectroscopy or DFS plots (for the other cases studied here) look?

-          Related to the above comment, can the authors show at least a comparative DFS plot for one of the virus strains and 2 cell models (for instance, for AH1 for both CHO and A549 cells)? The rest of the plots should be included in the supporting information.  

-          It would also be highly informative if the authors could discuss their data on cells in light of the interaction characteristics of isolated HA with isolated SA measured by single-molecule force spectroscopy.They could either include data or perhaps perhaps refer to the literature (see for example: Rabe et al., Biophysical Journal 116, 1037–1048, 2019 titled "Force Spectroscopy Shows Dynamic Binding of Influenza Hemagglutinin and Neuraminidase to Sialic Acid"), if relevant.

-          Besides calculating thermodynamics parameters, can authors also report on the mean unbinding forces for various virus strains and 2 cell models at a specific loading rate? In my opinion, this will provide additional information about the binding behaviour of various virus strains to cells.

-          The buffer conditions for virus binding on the glycan arrays and in SVFS appear to be different. In one case the buffer contains divalent ions (CaCl2 and MagCl2) in the other its only PBS. Is there a risk that the buffer conditions influence binding / binding specificity ?

-          Can the authors plot the dark and light gray histograms separately? In Figure 3B, the histogram value at ~25pN is not visible for the dark grey histogram (-NA).

-          While I generally agree with the authors that the reduction of binding probability and the unchanged binding forces suggest indeed that the virus interaction with the cell surface is mainly mediated by SA, why this is should be more clearly explained to the general readership of viruses.

Minor comments:

-          Check the reference (Figure 1c-f) in section 2.5 (line 15).

-          There might be a more appropriate location to cross reference Figure 2B than as done now on line 218 (Section 3.2).

-          In section 2.4, the information “experiment performed at room temperature” is mentioned twice.

-          There is a typo on line 320 “a an”.

Reviewer 2 Report

This manuscript investigates the interaction between various influenza A virus strains and distinct sialylated glycan types on host cell surfaces. The topic is important since HA recognition of different glycans plays a key role in interspecies transmission. Traditionally, such investigations relied on synthetic glycan arrays due to the heterogeneity and complexity of cell surface glycans. This study introduces an innovative approach, utilizing AFM to probe the interaction between individual virus and the host cell surface. However, while the technique is good, the paper falls short of drawing significant conclusions regarding the correlation between host cell surface glycan structure and cell specificity or the conclusion cannot be strongly supported by the results. 

Major concerns:

  1. 1 The study primarily biophysically measures binding, but actual infection also involves HA conformational changes and membrane fusion. The study design, where a single virus is attached to an AFM tip, seems unlikely to affect this process. Can the authors employ AFM to observe viral-host cell membrane fusion directly, rather than just binding?

  1. 2 Although the use of actual cells over synthetic glycans means better physiological relevance, the authors solely apply AFM. Complementary assays, such as in vitro pseudovirus assays employing luciferase as a reporter gene, should be included. The discrepancy between SVFS results and synthetic glycan assay is not unexpected given their inherent differences, yet the manuscript does not address this inconsistency specifically, leaving an intriguing question unanswered.
  2.  
  3. 3 The idea of non-sialic acid receptors contributing to HA-cell binding is bold and potentially groundbreaking. However, the SVFS results presented in this paper are insufficient to strongly support such a transformative hypothesis. The claim that NA treatment reduces binding probability but not unbinding force suggests the involvement of other cell surface molecules in determining cell specificity. Yet, this might also reflect the experimental setup's potential limitations concerning AFM specificity. Could the authors explore additional experimental approaches to corroborate this hypothesis? Or is there any other references can support this conclusion?
  4.  

Minor issues:

  1. the authors should confirm the covalent attachment of a single virus to the AFM tip.
  2. It could be the formatting but I could not find figure legends in the manuscript?
  3. The binding probability of 29.7% is relatively low, and even after NA treatment, a 5-13% binding probability remains. Additionally, the peak position of the pdf remains unaltered. This raises questions regarding the SVFS specificity. Can the authors elaborate on their conclusion that an unchanged pdf peak position implies good specificity (line 225-228)?
  4. Citations, particularly in the introduction section (e.g., line 45, 47, 50, 54, 63, etc.), appear not well formatted.

In summary, this manuscript offers an intriguing, promising, and information-rich method – SVFS, which can directly measure virus-cell interaction and provide insights into the energy barrier, dissociation rate, and bond life span. However, the evidence presented in the study does not robustly support the major conclusions. Further experimental data and additional analyses are required to support the claims made.

Round 2

Reviewer 1 Report

The authors have generally addressed all raised points in a satisfactory manner. However, concerning point 14, it is a pity that the authors do not comment further on the data in the main text, as it might be possible to draw some interesting information from  a comparison of the unbinding forces  (e.g. comparing how different viruses behave on different cells)  or by looking at the relative qualitative differences in affinities or binding forces for different virus pairs.

Two minor corrections that I noticed on the way:

Line 227: it is 4um/s instead of 4nm/s (as stated in Fig. S4)

Figure S2 contains a virus named Ro8. This one is not mentioned in the main text?

Author Response

Point 1: The authors have generally addressed all raised points in a satisfactory manner. However, concerning point 14, it is a pity that the authors do not comment further on the data in the main text, as it might be possible to draw some interesting information from a comparison of the unbinding forces (e.g. comparing how different viruses behave on different cells) or by looking at the relative qualitative differences in affinities or binding forces for different virus pairs.

Response 1: We have added a short comment on the unbinding force comparison shown in Figure S4 into the main text.

While we found the lowest mean unbinding force for AH1, pdmH1N1 and H7N1/FPV show unbinding forces 4-6 times larger, possibly indicating an affinity difference. The high unbinding forces are in agreement with lower off rates for pdmH1N1 and H7N1/FPV compared to the other virus-cell pairs.

Two minor corrections that I noticed on the way:

Point 2: Line 227: it is 4um/s instead of 4nm/s (as stated in Fig. S4)

Response 2: The error was corrected. Thank you.

Point 3: Figure S2 contains a virus named Ro8. This one is not mentioned in the main text?

Response 3: Thank you for pointing this out. We have updated Figure S2, now showing the same virus strain names as used in the main text.